# Few layer 2D pnictogens catalyze the alkylation of soft nucleophiles with esters

Vicent Lloret[1], Miguel Ángel Rivero-Crespo[2], José Alejandro Vidal-Moya[2], Stefan Wild[1],
Antonio Doménech-Carbó[3], Bettina S.J. Heller[4], Sunghwan Shin[4], Hans-Peter Steinrück[4], Florian Maier[4],
Frank Hauke[1], Maria Varela[5], Andreas Hirsch [1], Antonio Leyva-Pérez[2] & Gonzalo Abellán[1,6]

Group 15 elements in zero oxidation state (P, As, Sb and Bi), also called pnictogens, are rarely used in catalysis due to the difficulties associated in preparing well–structured and stable materials. Here, we report on the synthesis of highly exfoliated, few layer 2D phosphorene and antimonene in zero oxidation state, suspended in an ionic liquid, with the native atoms ready to interact with external reagents while avoiding aerobic or aqueous decomposition pathways, and on their use as efficient catalysts for the alkylation of nucleophiles with esters. The few layer pnictogen material circumvents the extremely harsh reaction conditions associated to previous superacid–catalyzed alkylations, by enabling an alternative mechanism on surface, protected from the water and air by the ionic liquid. These 2D catalysts allow the alkylation of a variety of acid–sensitive organic molecules and giving synthetic relevancy to the use of simple esters as alkylating agents.

[1] Department of Chemistry and Pharmacy & Joint Institute of Advanced Materials and Processes (ZMP), Friedrich-Alexander-Universität Erlangen-Nürnberg (FAU), Dr.-Mack-Straße 81, 90762 Fürth, Germany. [2] Instituto de Tecnología Química, Universidad Politécnica de Valencia, Consejo Superior de Investigaciones Científicas, Avda. de los Naranjos s/n, 46022 Valencia, Spain. [3] Departamento de Química Analítica, Universidad de Valencia, Dr. Moliner, 50, 46100 Burjassot, Valencia, Spain. [4] Chair of Physical Chemistry II, Friedrich-Alexander-Universität Erlangen-Nürnberg (FAU), Egerlandstr. 3, 91058 Erlangen, Germany. [5] Departamento de Física de Materiales, Universidad Complutense de Madrid, Instituto Pluridisciplinar, Instituto de Magnetismo Aplicado, Madrid 28040, Spain. [6] Instituto de Ciencia Molecular (ICMol), Universidad de Valencia, Catedrático José Beltrán 2, 46980 Paterna, Valencia, Spain. These authors contributed equally: Vicent Lloret, Miguel Ángel Rivero-Crespo. Correspondence and requests for materials should be addressed to A.L.-P. (email: anleyva@itq.upv.es) or to G.A. (email: gonzalo.abellan@fau.de)

Two-dimensional (2D) materials have attracted great attention in the last years due to their outstanding physical properties and their potential applications in optoelectronics, sensors, energy storage, and catalysis[1]. In contrast to the most studied material graphene, the layered allotropes of group 15 elements (P, As, Sb, and Bi, also called pnictogens) have been fairly less developed. 2D pnictogens exhibit a marked puckered structure[2–4] with dative electron lone pairs located on the surface atoms, which results in semiconducting character and good electronic mobility[4,5], and also in the ability to easily adsorb and stabilize, particularly well, unsaturated organic molecules through van der Waals interactions[6,7]. Thus, 2D pnictogens might, in principle, act as catalysts in synthetic organic transformations involving unsaturated molecules, in a completely different way as graphene does[8,9]. This concept, however, requires a new methodology to synthesize large amounts of exfoliated material, thus exposing most of the catalysts atoms to the outer space for maximizing interaction with substrate molecules.

Alkylation reactions are fundamental in biochemistry and organic synthesis. Nature makes use of alkyl phosphates, sulphonates, and esters as alkylating agents, under metal-free physiological conditions[10,11]. In contrast, synthetic methods generally employ energetically higher alkyl halides and alcohols as alkylating agents under very strong basic or acidic conditions (i.e., Williamson synthesis)[12], and the synthetic alkylation protocols reported with poly-oxygenated compounds need expensive and toxic metal catalysts, such as the palladium-catalyzed Tsuji–Trost allylation reaction[13], the Hantzsch ester-assisted hafnium-catalyzed alkylation of quinones[14], and the gold-catalyzed alkylation with alkynylbenzoic acids[15], among some others[16,17]. Thus, the discovery of a simple, metal-free, biomimetic alkylation reaction with readily available poly-oxygenated molecules[18] remains a challenge in organic synthesis and catalysis, furthermore attractive if selective and functional-group tolerant[19,20].

Here, we show the synthesis of two different exfoliated, few layer 2D pnictogens, phosphorene (few-layer black phosphorous (FL-BP)) and antimonene (FL-Sb), and their use as catalysts in the alkylation of alcohols, thiols, and indoles with simple esters, in good yields and with excellent selectivity. To our knowledge, this is the first organic reaction catalyzed by pristine 2D-pnictogens reported so far. Mechanistic studies unveil that the catalytic FL pnictogen selectively adsorbs the nucleophile and ester on surface, with the help of the electronic stabilization generated by the few layers underneath. FL-Sb exhibits a better performance than FL-BP, in accordance with its higher polarizability, enabling acid-sensitive aromatic derivatives to be selectively alkylated with simple esters.

## Results and Discussion

### Synthesis and characterization of FL-BP and FL-Sb in bmim–BF$_4$.

Figures 1a and 2a show the structure of FL-BP and FL-Sb nanosheets, respectively, produced by liquid phase exfoliation (LPE)[21,22]. This technique is often carried out in amide solvents such as N-cyclohexyl-2-pyrrolidone (CHP) or N-methyl-2-pyrrolidone (NMP). Here, the ionic liquid (IL) 1-butyl-3-methylimidazolium tetrafluoroborate (bmim-BF$_4$) is used on the basis of its excellent oxidation protection behavior for FL-BP[23]. Sonication of ground BP or Sb crystals dispersed in bmim-BF$_4$ was performed in an argon-filled glovebox (<0.1 ppm of $H_2O$ and $O_2$) to yield brownish, open-air stable suspensions of unoxidized FL-BP or Sb nanosheets, after removing the unexfoliated particles by a two-step centrifugation process, 14,000$g$ during 1 min, and then at 2000 and 100$g$ for 60 min for FL-BP and FL-Sb, respectively. The samples were stored under ambient conditions over weeks with no signature of degradation. In order to provide statistical information of the thicknesses and lateral dimensions of the as-prepared nanosheets, topographic atomic force microscopy (AFM) characterization, and spectroscopic micro-Raman mapping of >150 nanosheets, spin-coated onto SiO$_2$/Si wafers, was performed. The results showed that the BP particles have median values of ca. 150 nm in lateral dimensions and average thicknesses of 13 nm, with thinner particles down to a few nanometers being predominant (see Supplementary Figures 1–7). Particles smaller than 2 nm were excluded from statistics since capillary and adhesion effects of the IL account for average motifs of ca. 60 nm in lateral dimensions and ca. 1.8 nm in thickness (see Supplementary Figures 8 and 9)[24]. It is worth noting the general difficulties associated to AFM measurements in the presence of ILs, due to the high viscosity, adhesion forces, and formation of IL aggregates. The corresponding scanning Raman microscopy (SRM) spectra (>14,000 single point spectra), with an excitation wavelength of 532 nm, unambiguously showed the characteristic modes of BP, labeled A$_g^1$, B$_{2g}$, and A$_g^2$, with no signature of oxidation attending to the A$_g^1$/A$_g^2$ > 0.6 intensity ratio statistics, independent of the orientation (see Fig. 1 and Supplementary Figures 1–7 in Supplementary Information for additional Raman and AFM characterization)[23,25]. Aberration corrected scanning transmission electron microscopy (STEM) combined with electron energy-loss spectroscopy (EELS) was used to investigate the local structure and chemistry of the flakes. An atomic-resolution high-angle annular dark field (HAADF) STEM image of the FL-BP sample acquired down the [110] crystallographic direction, with the electron beam perpendicular to the platelet plane, is shown in Fig. 1g (both raw and Fourier filtered data, Supplementary Figure 10 displays a low magnification image of the flake). The samples exhibit a very high degree of crystallinity, showing the characteristic puckered structure over regions of hundreds of nanometers. The lattice shows high uniformity with the presence of very few defects or dislocations. Analysis by EELS shows that the IL locates and nanometrically covers the edges of the 2D material, as assessed by the P $L_{2,3}$–edges, the C K-edge, the N K-edge, and the O K-edge chemical maps in Fig. 1h, with onsets near 132, 284, 401, and 532 eV, respectively. X-ray diffraction (XRD) of a FL-BP sample, measured after washing with tetrahydrofurane under nitrogen atmosphere, ultracentrifugation and evaporation of the solvent, shows a spectrum consistent with BP, with the typical 020, 040, and 060 planes and without any sign of degradation nor oxidation, and when the sample was exposed to the ambient, the peak intensities rapidly decreased (Supplementary Figure 10). These results infer the high surface area of the FL-BP synthesized here.

To confirm the zero oxidation state of P and rule out partial reduction of oxidized P species in EELS by the electron beam, X-ray photoelectron spectroscopy (XPS) studies have been carried out under ultra-high vacuum (UHV) conditions on highly concentrated IL FL-BP suspensions (FL-BP$_{sus}$) coating a clean Au foil as support. The overview spectrum of FL-BP$_{sus}$ shows the expected IL core levels (Supplementary Figure 11) and, additionally, typical Si/O/C signals of trace bulk contamination after contact with glassware grease, due to surface enrichment effects as it is often the case in XP spectra for IL systems[26]. At around 130 eV binding energy, a small signal of the spin-orbit split P $2p_{1/2,3/2}$ signal is detected that is absent for the neat IL (Fig. 1i, spectra II and III). The binding energy position of the P $2p_{3/2}$ level at 130.2 eV can be unambiguously assigned to BP in oxidation state zero and the absence of signals between 132 and 137 eV rules out significant oxidized P species being present[27]. By heating the FL-BP$_{sus}$ sample above 150 °C for 1 h in UHV, most of the IL was gone by thermal desorption (and partial decomposition), which led to an increase of the BP signal intensity as dominating remaining species by a factor around 50 (Fig. 1i, spectrum III); again, no oxidized P species could be

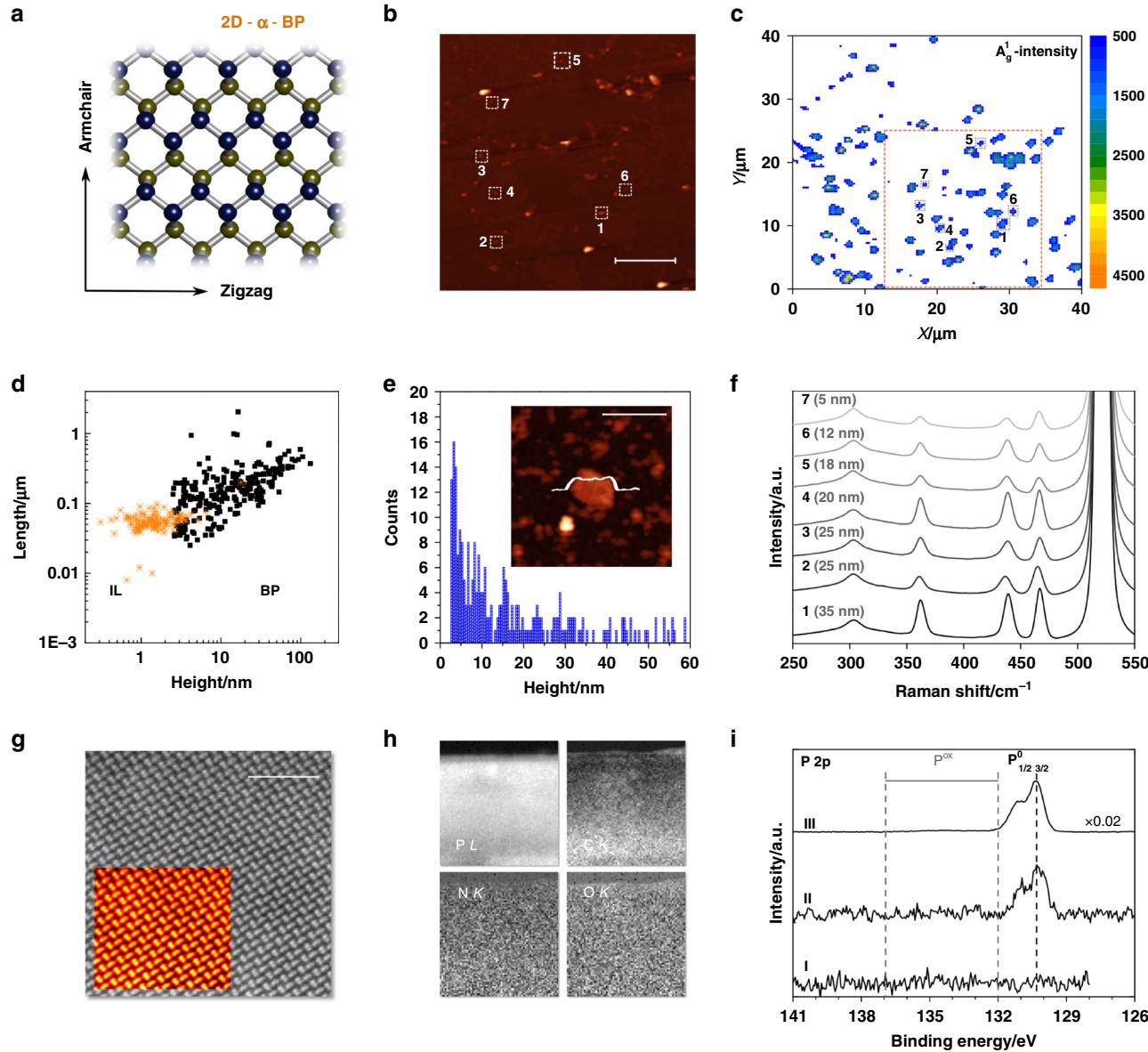

**Fig. 1** Characterization of exfoliated black phosphorus. **a** Top view of the orthorhombic structure ($C_{mce}$ space group) of BP. Upper plane P atoms marked in blue, lower plane in yellow. **b** Representative AFM topography image (inset, scale bar 5 µm) of the exfoliated sample spin-coated onto $SiO_2$/Si substrates. **c** The corresponding Raman $A^1_g$ ($\lambda_{exc} = 532$ nm) mapping of the same BP flakes (>14,000 single point spectra over a surface area of 40µm² using a step size of 0.5 µm). The numbers denote the areas in which the Raman spectra shown in (**f**) were recorded. **d** Plot of the nanosheet length as a function of the flake height (obtained from AFM) considering a total amount of 252 replicates as well as the data corresponding to the IL blank (sample size = 116) Supporting Information S8 and 9). **e** Histogram of the apparent thickness of the exfoliated FL-BP obtained from AFM (sample size = 252). The inset shows an AFM image of a nanosheet along with its corresponding height profile of *ca.* 6 nm (inset, scale bar 400 nm). **f** Raman spectra of the nanosheets indicated in **b** and **c** (the numbers labeling the spectra in **f** correspond to the nanosheets marked by numbers in **b** and **c**). **g** Atomic resolution HAADF image acquired down the [110] axis, from the edge of a free-standing portion of a flake. The inset exhibits the raw image and a Fourier filtered (FFT) version, in false color. The scale bar represents 2 nm. **h** Compositional maps derived from electron energy-loss spectroscopy (EELS) measurements acquired on the free-standing portion of the BP flake. The P $L_{2,3}$, C $K$, N $K$, and O $K$ maps for this area are shown, the scale bar represents 20 nm. Data acquired at 80 kV. **i** XPS P 2p region of the neat bmim-BF₄ IL (I), the highly concentrated FL-BP suspension (II) showing only P in oxidation state zero at P $2p_{3/2} = 130.2$ eV (region for oxidized P species is indicated), and after removal of most of the IL by heating in UHV (III); spectra are offset and rescaled for sake of clarity. Source data are provided as a Source Data file

detected. In order to check if the heated FL-BP$_{sus}$ sample with most of the protecting IL removed was now prone to oxidation, this sample was subsequently exposed to air under ambient conditions for about one day and measured again. XPS clearly revealed a broad oxide P component around 134 eV (Supplementary Figure 12, spectrum IV) as has already been observed for in situ oxidation studies of BP[27].

In the case of FL-Sb, the shorter out-of-plane atom-to-atom distances, which are indicative of stronger interlayer interactions, usually hampers mechanical exfoliation. However, the LPE approach here used was able to give median values of 310 nm in lateral dimensions and *ca.* 32 nm in thickness (extracted from >150 flakes), as it can be observed in Fig. 2 and Supplementary Figures 13–18, with a minimum observed apparent thickness of 4 nm[22]. The SRM

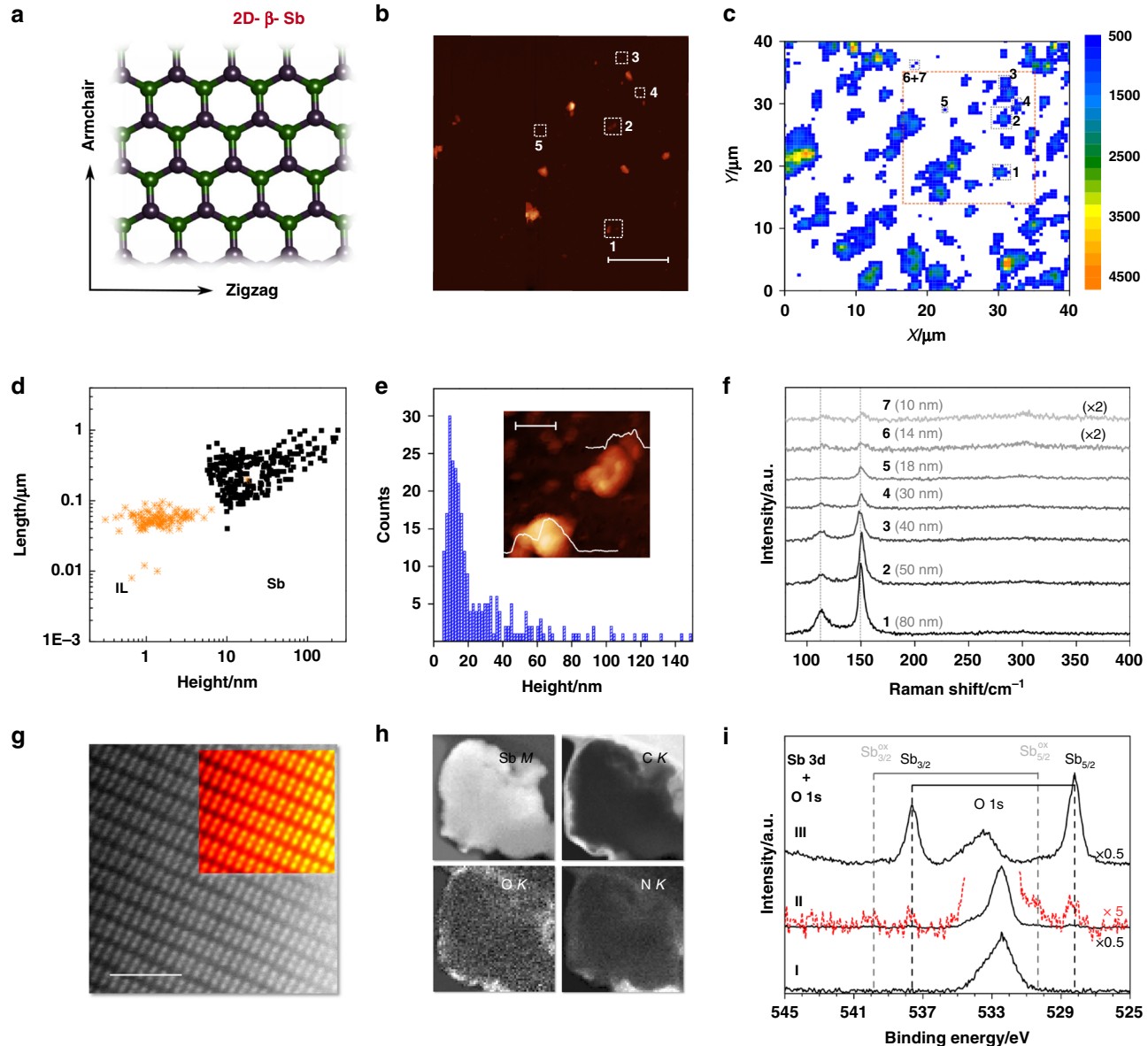

**Fig. 2** Characterization of exfoliated antimonene. FL-Sb analysis. **a** Top view of the rhombohedral structure (R3m space group) of Sb. Upper plane atoms marked in green, lower plane in purple. **b** Representative AFM topography image (inset, scale bar 5 μm) of the exfoliated sample spin-coated onto SiO$_2$/Si substrates. **c** The corresponding Raman A$^1_g$ ($\lambda_{exc}$ = 532 nm) mapping of the same Sb flakes (>14,000 single point spectra over a surface area of 40 μm$^2$ using a step size of 0.5 μm). The numbers denote the areas in which the Raman spectra shown in (**f**) were recorded. **d** Plot of the nanosheet length as a function of the flake height (obtained from AFM), including the data corresponding to the IL blank (Supplementary Figures 8 and 9) considering a total amount of 271 and 116 replicates, respectively. **e** Histogram of the apparent thickness of the exfoliated FL-Sb obtained from AFM (sample size = 271). The inset shows an AFM image of a nanosheet along with its corresponding height profiles of *ca.* 4 and 18 nm, respectively (inset, scale bar 100 nm). **f** Raman spectra of the nanosheets indicated in **b** and **c** (the numbers labeling the spectra in **f** correspond to the nanosheets marked by numbers in **b** and **c**). **g** Atomic resolution HAADF image acquired on the edge of a free-standing portion of a flake, near the edge, along with a Fourier filtered (FFT) image in the inset, acquired down the [210] orientation. The scale bar is 2 nm. **h** Compositional maps, derived from EEL spectrum images of the flake (from the area highlighted with a green rectangle in Supplementary Figure 21). Sb $M_{4,5}$, C $K$, O $K$, and N $K$ maps corresponding to this area are shown. Data acquired at 80 kV. **i** XPS Sb 3$d$ and O 1$s$ region of the neat bmim-BF$_4$ IL (I) showing oxygen signals from an IL related surface contamination layer, of the highly concentrated FL-Sb suspension (II) showing small signals of nonoxidized (Sb 3$d_{5/2}$ at 528.2 eV) and minor contributions from oxidized (530.3 eV) antimony next to the oxygen contamination, and after removal of most of the IL by heating in UHV (III); spectra are offset and rescaled for sake of clarity. Source data are provided as a Source Data file

mappings revealed the characteristic main phonon peaks, the A$^1_g$ mode at 149.8 cm$^{-1}$ and E$_g$ mode at 110 cm$^{-1}$, even for the thinnest particles with no signature of oxidation (peaks related to the formation of Sb$_2$O$_3$ or Sb$_2$O$_5$). A phonon softening effect (blueshift) was observed when the sample thickness decreases from the bulk to *ca.* 10 nm, in good agreement with theoretical predictions and recent reports (Fig. 2f and Supplementary Figures 19 and 20)[4,22,28,29]. The E$_g$/A$^1_g$ intensity ratio (measured using 532 nm excitation wavelength) increases from 0.37 to 0.79 with thickness decreasing from 80 to 10 nm (Supplementary Figure 19)[29].

FL-Sb electron microscopy images denote irregularly shaped submicrometric flakes, with lateral sizes in the range of hundreds

of nm, as assessed by a low-magnification HAADF image of a flake and the atomic-resolution image of the crystal structure (Fig. 2g and Supplementary 21), both obtained at an acceleration voltage of 80 kV to prevent beam-induced damage. This structure agrees with that of β-antimony along the [2 1 0] direction. Again, the samples are highly crystalline and no major defects were observed. EELS maps exhibit a C-rich coating consisting of an amorphous layer a few nanometers thick, as well as the presence of O, N, and C mostly located around the edges (Fig. 2h). The seemingly preferential location of the IL molecules along the edges is in good agreement with the expected higher polarity of the unsaturated atoms of the 2D material[6,23,30].

As done for the FL-BP system, highly concentrated IL FL-Sb suspensions (FL-Sb$_{sus}$) were investigated using XPS. Next to the broad O 1$s$ signal at 533 eV from the surface enriched IL contamination (see also overview spectrum shown in Supplementary Figure 22), the Sb 3$d$ region between 525 and 545 eV of the FL-Sb$_{sus}$ sample (Fig. 2i, spectrum II) reveals very weak Sb 3$d_{3/2,5/2}$ signals from antimony in oxidation state zero at 528.2 eV for the 3$d_{5/2}$ level, along with minor contributions from Sb in higher oxidation state at around 530.3 eV (Fig. 2i, magnified red spectrum II)[31,32]. Removing most of the excess IL by heating in UHV clearly showed Sb signals originating mostly from bulk antimony zero (Fig. 2i, spectrum III). Exposing the heated FL-Sb$_{sus}$ sample without the protecting IL medium for several hours to air led to a drastic decrease in Sb(0) and concomitant increase of the oxidized Sb species (Supplementary Figure 23, spectrum IV); these findings thus strongly supports the role of bmim-BF$_4$ stabilizing the P and Sb pnictogens in IL solution against oxidation.

**Catalytic alkylation with esters.** Substitution reactions are fundamental transformations in organic chemistry that, due to their bimolecular nature and in order to be performed selectively, require the use of a catalyst able to activate the substrates orthogonally. Highly polarized Lewis bases are, in principle, suitable species to carry out a bimolecular and orthogonal catalytic activation, and FL-BP and FL-Sb may act in this way due to the intrinsic electron richness of the bulk atoms (base) combined with the expected stabilization of the in situ generated cationic charge.

As a reaction proof, the *tert*-butylation of alcohols, a longstanding challenge in organic synthesis[33], was studied. Current methodologies at laboratory[34] or industrial scale[35] for this reaction are still based on Friedel–Crafts type chemistry, with isobutylene or *tert*-butyl alcohol as alkylating agents under very strong reaction conditions. These harsh protocols are substrate-limiting and particularly unselective in the presence of aromatic rings[36], despite synthetically elaborated, energetically richer and expensive *tert*-butylated reagents have been prepared on purpose to mitigate these drawbacks[37].

Table 1 shows the results for the reaction between benzyl alcohol **1** and *tert*-butyl acetate **2**, in the presence of different catalysts. FL-BP and FL-Sb exclusively gives *tert*-butyl ether **3** after 20 h at 75 °C, in reasonable yields and with >99% selectivity (the rest is unreacted material, entries 2 and 3). In contrast, other bifunctional layered materials (entries 4–7) and semiconductors with high surface area (entries 8–10), and also inorganic and organic bases of different strength (entries 11–18), do not show any significant catalytic activity under these reaction conditions.

Isotopic experiments with $^{18}$O-**1** confirm that the oxygen atom of the alcohol stays intact in product **3**, which suggests that an ester C–O alkyl cleavage (AL mechanism) operates, as reported with superacids ($H_0 < 0$) such as HSO$_3$F–SbF$_5$–SO$_2$[33]. Notice that, for any weaker acid, the AL mechanism is rapidly undertook by the more common acyl cleavage (AC) mechanism, to give the

trans-esterification reaction. Indeed, acetic acid (AcOH) is too weak to catalyze the reaction between **1** and **2** (entry 19), sulfuric and triflimidic acid show moderate catalytic activity and give ester **3'** as the major product (entries 20 and 21), and only the superacid triflic acid (HOTf) shows catalytic activity but with still poor selectivity towards **3** (entry 22). The increase in selectivity for **3** with acid strength is in good agreement with the need of forming the carbocation intermediate of the AL mechanism, to trigger the unimolecular A$_{AL}$1 reaction and give ether **3**. Strong Lewis acids were also tried (entries 23–32), and while some of them showed some activity (entries 24–25, 29, and 31–32), their activity corresponds exclusively to the Brönsted acidity of the in situ hydrolyzed anions, as confirmed by the lack of activity when the proton quencher 2,6-di*tert*-butylpyridine is used. Notice that the harsh reaction conditions associated with superacids are incompatible with most functional organic groups, thus any other nucleophile beyond water has not regularly been employed for the superacid-catalyzed alkylation with esters, as far as we know. The initial turnover frequency (TOF$_0$) for FL-BP and FL-Sb (20 and 73 h$^{-1}$, respectively) are in the range of the strong acids (between 8 and 155 h$^{-1}$), which reflects the good intrinsic activity per atom of the pnictogen 2D materials. Strong Brönsted solid acids such as sulphated zirconia and silica–alumina gave only a marginal catalytic activity.

Figure 3 shows the scope for the FL-Sb catalyzed reaction, with different nucleophiles and esters. The results show that a variety of *tert*-butyl esters give ether **3** in >70% yield, up to 2-gram scale, including unsaturated esters and carbonates, and that benzylic (products **4** and **10**) and alkyl alcohols (products **5** and **11**), thiols (products **6–7**), indoles, either in the carbon (products **8a–9a**) or nitrogen atom (products **8b–9b**), and phenols (product **11**) can be alkylated with acetates having either *tert*–butyl, cinnamyl, benzyl, and prenyl moieties (in blue). Acid-sensitive functionalities are tolerated under the reaction conditions, such as ether, prenyl, trifluoroacetate and lactone groups (in red). The uniqueness of this synthetic approach is illustrated, for instance, for allylic benzylic alcohols, since product **10** has not been synthesized so far and none of the nine methods reported previously for the generic structure, according to a literature searching, provides a so simple, direct and efficient method as 2D-pnictogens do (Table S1).

Recovery tests (Supplementary Figure 24) show that FL–BP and FL-Sb can be reused at least three times under ambient conditions before deactivation occurs. STEM analysis of the reused samples shows a progressive amorphization of the edges (see Supplementary Figure 25 for further details), which is significantly alleviated by reusing the material under N$_2$ atmosphere, thus prolonging catalyst lifetime[23,38]. In order to further assess the stability of the material *in operando* conditions, $^{31}$P magic angle spinning nuclear magnetic resonance (MAS NMR) spectra of the FL-BP catalysts were recorded under reaction conditions. The results (Supplementary Figure 26, top) show that the FL-BP catalyst keeps the original signal at 18 ppm during reaction[39], without any trace of phosphoric acid (0 ppm), phosphonium compounds, or other potential oxidized and hydrolyzed species. The measurement was also performed in statics (Supplementary Figure 26, bottom), and the deconvoluted spectrum fits to two components, one corresponding to P in zero oxidation state and the other corresponding to P–O, which accounts for 8% out of the total. Electrochemistry measurements (Supplementary Figure 27) strongly supports the absence of neat oxidation P or Sb species during reaction[40], since no significant changes in the 2D pnictogen signals were found, and if any, they correlate with particle aggregation/fractioning processes rather than the formation of phosphonium or further oxidized P and Sb forms. Following this rationale, and in order to increase the

**Table 1 Results for the catalytic alkylation of 1 with 2. Conversion of 1 is the sum of 3 and 3′ yields, the rest is unreacted material**

Reaction scheme: Benzyl alcohol (1, with $^{18}O$) + $Me-C(=O)-O-{}^tBu$ (2, 1.5 equiv.) → Catalyst (5 mol%), bmim-BF$_4$ (1 M), 75 °C, 20 h → 3 (AL mechanism) (-AcOH) + 3′ (AC mechanism) (-$^t$BuOH)

| Entry | Catalyst | TOF$_0$ (h$^{-1}$) | Yield of 3 (%) | Yield of 3′ (%) | Selectivity AL/AC (%) |
|---|---|---|---|---|---|
| 1 | None | — | — | — | — |
| 2 | **Phosphorene (FL-BP)** | **20.1** | **46** | <1 | >99 |
| 3 | **Antimonene (FL-Sb)** | **73.2** | **57** | <1 | >99 |
| 4 | Graphene | — | 4 | <1 | — |
| 5 | Boron nitride | — | 3 | <1 | — |
| 6 | Hydrotalcite | — | <1 | <1 | — |
| 7 | Sepiolite | — | 6 | <1 | — |
| 8 | Nano–TiO$_2$ | — | 2 | <1 | — |
| 9 | Nano–CeO$_2$ | — | <1 | <1 | — |
| 10 | Nano–Fe$_2$O$_3$ | — | <1 | <1 | — |
| 11 | K$_3$PO$_4$ | — | <1 | <1 | — |
| 12 | Na$_2$CO$_3$ | — | <1 | <1 | — |
| 13 | Pyridine | — | <1 | <1 | — |
| 14 | Et$_3$N | — | <1 | <1 | — |
| 15 | KOAc | — | <1 | <1 | — |
| 16 | KO$^t$Bu | — | <1 | <1 | — |
| 17 | DABCO | — | <1 | <1 | — |
| 18 | NanoMgO | — | <1 | <1 | — |
| 19 | HOAc | — | <1 | <1 | — |
| 20 | H$_2$SO$_4$ | 23.7 | 24 | 46 | 35 |
| 21 | HNTf$_2$ | 7.6 | 8 | 14 | 25 |
| 22 | HOTf | 154.2 | 48 | 52 | 48 |
| 23 | MnOAc$_2$ | — | 0 | — | — |
| 24[a] | FeCl$_2$ | 2.2 | 33 | <1 | >99 |
| 25[a] | CoCl$_2$ | 7.4 | 44 | <1 | >99 |
| 26 | CuCl | — | 2 | — | — |
| 27 | AgMeSO$_4$ | — | 2 | <1 | — |
| 28 | Pd(OAc)$_2$ | — | 6 | — | — |
| 29[a] | PtCl$_2$ | — | 25 | <1 | >99 |
| 30 | Au(OH)$_3$ | — | 0 | — | — |
| 31[a] | Bi(OTf)$_3$ | 6.0 | 37 | 10 | 79 |
| 32[a] | CeCl$_3$ | 7.0 | 52 | <1 | >99 |
| 33 | Sulphated zirconia | — | 11 | <1 | >99 |
| 34 | SiO$_2$–Al$_2$O$_3$ (13%) | — | 0 | — | — |

[a]No catalytic activity if 2,6-di*tert*-butylpyridine (30 mol%) is added during reaction

catalyst lifetime, sonication of the IL mixture was carried out before each reuse, and in this way, only a minor loss of final yield was observed after six reuses, thus doubling the catalytic performance without sonication.

Figure 4a shows the rate equation of the reaction between **1** and **2** derived from kinetic experiments (Supplementary Figure 28), with either FL-BP or FL-Sb catalyst. The results give $v_0 = K_{app1}[\text{P or Sb}][\mathbf{1}][\mathbf{2}]$, that indicates that the three species, i.e., the alcohol, the ester and the pnictogen catalyst, are involved in the rate-determining step of the reaction. This equation rate differs significantly from a classical A$_{AL}$1 mechanism[33,41,42], where the nucleophile does not participate in the rate-limiting step since only carbocation formation controls the overall reaction rate. Accordingly, kinetic experiments with HOTf, under the reaction conditions employed here, show that **1** does not participate during the rate determining step, with an equation rate $v_0 = K_{app2}[\text{H}^+][\mathbf{2}]$. This result confirms that the classical A$_{AL}$1 mechanism operates with HOTf in BMIM-BF$_4$ (Supplementary Figure 29), and that it is different from that with FL-BP or FL-Sb. Analysis of the reaction gas phase by gas chromatography coupled to mass spectrometry (GC–MS) shows that nearly 0.5 equivalents of **2** are transformed to isobutylene gas during the HOTf-catalyzed reaction, regardless if **1** is present or not, while only traces of isobutylene gas are detected with the FL-BP catalyst. Control experiments with isobutylene or styrene as reagents, instead of the corresponding esters, discard alkenes as

**Fig. 3** Substrate scope for the FL-Sb catalyzed alkylation with esters. R[1] is Me (acetate) unless otherwise indicated. In blue, the introduced alkyl moiety; in red, particularly acid–sensitive functional groups. **a** 2-gram scale. **b** 25 mol% catalyst, 48 h. **c** Three equivalents of ester

alkylating agents under the present reaction conditions. Notice that isobutylene is a typical by-product of long-lived *tert*–butyl cations, and the preferential formation with HOTf and the different equation rates illustrate the striking differences between the FL-pnictogen and the superacid catalyst mechanisms.

To further check the formation of carbocations or not during 2D pnictogen catalysis, *R*-1-phenylethanol acetate **13** was used as the alkylating agent for methanol. The results in Fig. 4b show that a racemic mixture of the alkylated product 1-phenylethyl methyl ether **14a**, together with isomerically pure 1-phenylethanol **14b** and starting **13**, were found as main products with FL-BP catalyst. This result unambiguously demonstrates that the alkyl moiety is transformed on the 2D pnictogen surface into a carbocation, at some point before transferring, since it is able to racemize prior to nucleophile addition. Of course, this also occurs with HOTf catalyst; however, the different equation rates and isobutylene yield found for the 2D pnictogen and HOTf catalysts suggest different carbocation managing during the alkylation reaction. Released acetic acid or traces of water do not act as nucleophiles towards the carbocation in both cases.

Figure 4c shows competitive tests between benzyl alcohol **1** and decyl alcohol **15** with **2**. The relative initial rates and final yields differ dramatically for FL-BP and HOTf; while FL-BP shows one order of magnitude (10.0) higher formation rate for the aromatic than for the alkyl alcohol, HOTf shows a relative rate of 1.7. These values give 6 times higher selectivity for aromatic substrates with FL-BP. Another clear difference is found in the corresponding Hammett plots: while electron withdrawing groups on the benzyl alcohol increase the reaction rate with the FL-BP catalyst, electron donor groups increase the rate with HOTf catalyst, the latter being the expected behavior for a free nucleophile (Supplementary Figure 30). Kinetic experiments with isotopically labeled D$_2$-**1** (PhCD$_2$OH) give a significant secondary kinetic

isotopic effect of 1.4(5) for the FL-BP catalyst. This last result, together with the Hammett plot, suggests an electron donation of the exfoliated material to the aromatic ring and then, in a lesser extent, to the alcohol by induction effects, in accordance to the known ability of 2D-BP[6] and 2D-Sb[7] to transfer electron charge to planar aromatic molecules.

Calorimetry measurements with **1** and FL-BP (Supplementary Figure 31) show that the aromatic alcohol strongly adsorbs to the BP surface, even at just 30 °C. Taft plots (Supplementary Figure 32) shows a clear positive slope for the 2D materials, which indicates that bulkiness on the aromatic ring diminishes reactivity. Decoupling steric and inductive effects by least-squares Taft regression analysis[43] (Tables S2–S4) confirms the need of co-planarity of the aromatic alcohol with the 2D catalyst, since the δ values obtained by the Taft regression analysis are the same than the Hammett plot, within the experimental error (−0.3 ± 0.5 for HOTf, 0.89 ± 0.08 FL-BP, and 0.5 ± 0.3 for Sb-BP). 2D nanosheets with different thickness were prepared by different ultra-centrifugation steps, with nearly one order of magnitude accessible bulk and edge atoms (Supplementary Figs. 33–38)[21], and kinetic results for the alkylation of **1** with **2** showed that the TOF$_0$ increases for the centrifuged samples. Since unsaturated P atoms in vertexes and edges are likely oxidized under reaction conditions and do not participate during catalysis, these results indicate that the catalysis is directly related to the number of atoms on the bulk (Supplementary Figure 39)[21,44]. When nitrobenzene was used as an inhibitor reagent, the alkylation rate of **1** with **2** was progressively quenched up to 20 mol% of nitrobenzene (respect to FL-BP, Supplementary Figure 40), which indicates that the high electron-deficient aromatic ring of nitrobenzene is strongly adsorbing to the bulk atoms of the 2D material. This inhibition value fits well to the number of P atoms present on

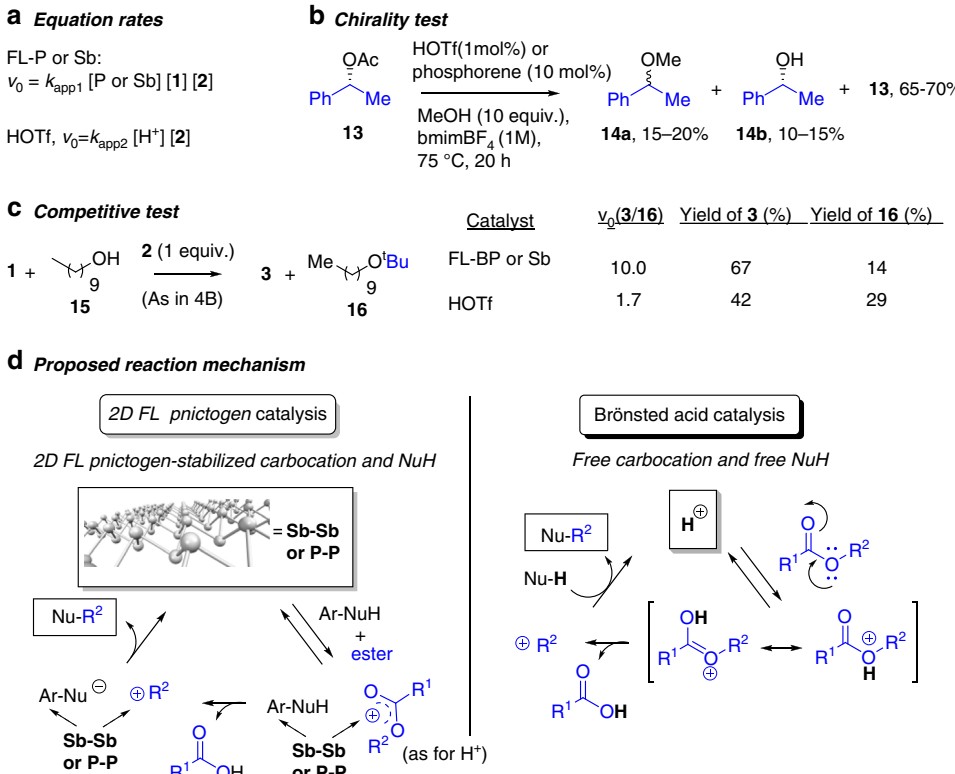

**Fig. 4** Mechanistic studies. Experimental evidences and proposed mechanisms for FL-BP, FL-Sb, and HOTf catalyzed-alkylation of nucleophiles with esters in bmim-BF$_4$

the whole bulk surface. These results point again to bulk P or Sb atoms as responsible for the catalysis.

P and Sb are very suitable atoms to stabilize carboxoniums, furthermore if the carbocation-oxonium equilibrium on surface is shifted toward the latter and assures a short living of the former. Indeed, low amounts of isobutylene were found during the alkylation reaction with FL-BP and FL-Sb, which supports a short living carbocation. A Hammett plot with different *para*-substituted *tert*-butyl benzoate esters shows that a positive charge is formed in the ester during reaction, thus carboxoniums can be presumed as intermediates (Supplementary Figure 41).

With all these data in hand, a very plausible mechanism for the 2D FL pnictogen-catalyzed alkylation of **1** with **2** is shown in Fig. 4d along the classical AL mechanism with HOTf for comparison, are shown in Fig. 4d. For the 2D FL pnictogen, the electron-rich 2D surface transfers electron density to the adsorbed aromatic nucleophile, which generates a deficiency of charge in the material stabilized by the few layers underneath[6,7] that activates the ester group and stabilizes a carboxonium inter-mediate, rapidly trapped by the nucleophile on surface and, thus, regenerating the electro-neutrality of the catalytic material. The higher polarizability of Sb vs. P explains the higher catalytic activity of the former. Notice that the activation of the ester occurs only after the bulk atoms act as a Lewis base, the natural behavior of 2D-BP and 2D-Sb, which enables a unique mechanism for the nanometric FL-BP and FL-Sb.

The transition state energetic values for FL-BP, calculated by an Eyring plot with the reaction between **1** and **2**, are $\Delta H^{\#} = 49.1$(4) kJ mol$^{-1}$ and $\Delta S^{\#} = 0.0(2)$ kJ·(mol K)$^{-1}$, and the unchanged entropy is consistent with both reactants binding to bulk P atoms, to couple rapidly after carbocation formation. In clear contrast, the mechanism for the superacid HOTf shows the formation of a free carbocation in solution, then trapped by the nucleophile. The

mechanisms proposed in Fig. 4d satisfactorily explain the experimental differences observed for the 2D-pnictogen and HOTf catalysts, and in particular, the equation rate, higher reactivity of EW aromatic nucleophiles and carbocation racemi-zation without alkene formation for the 2D-pnictogen catalyst. Other potential mechanisms for the alkylation reaction such as the formation of phosphonium intermediates, or neat redox processes on the pnictogen surface fostered by their narrow homo–lumo gap, seems not to be pointed out by the experimental evidences shown above.

The particular nature of the 2D-pnictogen materials described here, embedded in water- and oxygen-protecting ILs, and with a high basal area, allows organic catalysis, and not necessarily other expected applications for BP. For instance, water splitting under typical experimental conditions did not give any H$_2$ formation. The suitability of FL-BP and FL-Sb for organic catalysis is further supported by the preliminary positive results found for the catalytic Bailys–Hillman reaction (Supplementary Figure 42), a representative carbon–carbon bond-forming reaction.

2D-pnictogens widens and complements graphene catalysis (carbocatalysis), since when graphene is doped with precisely pinpointed heteroatoms, typically nitrogen, its catalytic activity equals simple metal catalysts (Table S5), which is not more what 2D-pnictogens make naturally here, without any additional modification. This rationale drives to think that 2D-pnictogens are potential advanced versions of doped graphene and extremely promising catalysts in organic reactions.

## Discussion

2D nanosheets of BP and Sb, exfoliated in the IL bmim-BF$_4$, catalyze the alkylation of soft nucleophiles with alkyl esters, in good yields and selectivity, particularly for aromatic substrates.

The 2D few layer materials circumvent superacid-mediated alkylations by enabling a surface mechanism with concomitant activation of the aromatic nucleophile and ester on surface, which allows acid-sensitive molecules to be alkylated. As far as we know, this is the first example of catalytic application of pristine FL-BP and FL-Sb in organic synthesis, beyond recent examples on photonic excitation or with supported metal nanoparticles[45–49], thus expanding the list of potential applications for these promising materials.

## Methods

**Materials and exfoliation process**. Throughout all experiments, BP and Sb with purity higher than 99.999% (Smart Elements) were used.

Afterwards, the FL-BP flakes were transferred onto Si/SiO$_2$ substrates (300 nm oxide layer). The exfoliation was performed in argon filled LABmasterpro sp glove box (MBraun) equipped with a gas purifier and solvent vapor removal unit (oxygen and water content lower than 0.1 ppm).

Solvent purification: anhydrous, 99.9% purity 1-butyl-3-methylimidazolium tetrafluoroborate (bmim-BF$_4$) was purchased from Sigma-Aldrich. The bmim-BF$_4$ was pump freezed, and the O$_2$ was removed by vacuum. This procedure was iteratively repeated a minimum of four cycles to remove traces of oxygen.

**Exfoliation of layered pnictogens**. LPE in bmim-BF$_4$ under inert conditions: BP was exfoliated under inert conditions by sonication in an argon-filled glovebox (O$_2$ < 0.1 ppm; H$_2$O < 0.1 ppm) using a Bandelin Sonoplus 3100, 25% amplitude, 12 h, pulse 2 s on, 2 s off. Sb was exfoliated under the same inert conditions by sonication using 40% amplitude, 16 h, pulse 2 s on, 2 s off. The starting concentrations were 1.25 and 2.5 mg mL$^{-1}$ for BP and Sb, respectively. The resultant dispersions were decanted and transferred into vials. All solvent transfer was carried out in the glovebox.

**Centrifugation**. Centrifugation was carried out in a MPW-350R centrifuge using 2 mL Eppendorfs. A two-step process was followed; first supernatant dispersion was centrifuged 14,000$g$ during 1 min, and afterwards the resulting supernatant was submitted to a second step at 2000 and 100$g$ for 60 min for FL-BP and FL-Sb, respectively. The final concentrations were determined by ICP-OES being $ca.$ 0.125 and 0.075 mg mL$^{-1}$ for FL-BP and FL-Sb, respectively. Longer centrifugations periods were performed to prepare samples of different thickness.

**Photoluminiscence**. PL of the different centrifuged samples was acquired on a Horiba Scientific Fluorolog-3 system equipped with 450 W Xe halogen lamp, double monochromator in excitation (grating 600 lines/mm blazed at 500 nm) and emission (grating 100 lines/mm blazed at 780 nm) and a nitrogen cooled InGaS diode array detector (Symphony iHR 320). Spectra were obtained at 5 °C measured (spectral region of 550–1300 nm) with a 550 nm cutoff filter in emission. Excitation and emission band widths were typically 10 nm and integration times 2 s.

**Surface preparation**. SiO$_2$ surfaces were sonicated for 15 min. in acetone and 15 min in 2-propanol and then dried under an argon flow.

Immediately after the removal from the inert atmosphere, images of FL-BP flakes were recorded under an optical microscope (Zeiss Axio Imager M1m), using different objectives enabling their relocalization in Raman and AFM measurements.

**Raman spectroscopy**. Raman spectra were acquired on a LabRam HR Evolution confocal Raman microscope (Horiba) equipped with an automated XYZ table using 0.80 NA objectives. All measurements were conducted using an excitation wavelength of 532 nm, with an acquisition time of 2 s and a grating of 1800 grooves/mm. To minimize the photo–induced laser oxidation of the samples, the laser intensity was kept at 5% (0.88 mW). The step sizes in the Raman mappings were in the 0.2–0.5 µm range depending on the experiments. Data processing was performed using Lab Spec 5 as evaluation software. When extracting mean intensities of individual BP Raman modes, it is important to keep each spectral range constant, e.g., from 355 to 370 cm$^{-1}$ and from 460 to 475 cm$^{-1}$ because otherwise the resulting value of the A$^1_g$/A$^2_g$-ratio can be slightly influenced. The same applies to Sb E$_g$/A$^1_g$ ratio analyses.

**Atomic force microscopy**. AFM was carried out using a Bruker Dimension Icon microscope in tapping–mode. The samples were prepared by spin coating a solution of a given sample at 5000 rpm. Bruker Scanasyst-Air silicon tips on nitride levers with a spring constant of 0.4 N m$^{-1}$ were used to obtain images resolved by 512 × 512 or 1024 × 1024 pixels.

**Scanning transmission electron microscopy**. STEM observations were carried out in a JEOL ARM200cF operated at 80 kV and equipped with a spherical aberration corrector and a Gatan Quantum EELS, at the ICTS-ELECMI Centro Nacional de Microscopía Electronica at UCM (Spain). Compositional maps were produced using a multiple linear least squares fit of the data to reference EEL spectra.

**X-ray photoelectron spectroscopy**. XPS measurements were carried out in an ultra-high vacuum (UHV, base pressure < 1 × 10$^{-10}$ mbar) system dedicated for angle-resolved XPS at UHV-compatible liquid samples such as ILs[50]. Spectra were collected in normal emission (information depth in ILs: 7–9 nm, depending on electron kinetic energy) using monochromated Al Kα radiation (1486.6 eV) and a pass energy of the hemispherical electron analyzer of 35 eV (overall energy resolution: 0.4 eV). Binding energies are referenced to C 1$s$ for aliphatic carbon of the IL (285.0 ± 0.1 eV) and Au 4$f_{7/2}$ (84.0 ± 0.1 eV) of clean gold. XPS-detection limit typically is around 1 at% depending on relative XPS cross-sections and signal-to-background situation for the trace atom[51]. The FL-Sb and FL-BP solutions employed for our catalysis studies with an overall P- and Sb-content around 0.1 mg/mL (that is, below 0.01 at% for P and Sb) were thus below the XPS detection limit as has been tested: only bmim-BF$_4$ signals could be detected along with a Si/O/C containing trace contamination that commonly shows up in I-XPS investigations due to surface enrichment effects, and is typically due to contact with glassware grease[26]. Hence, highly concentrated suspensions (2D-inks) of FL-BP and FL-Sb were prepared by filtering the dispersions through a 0.2 µm reinforced cellulose membrane filter (Sartorius) in the glove box to remove most of the IL. The filtration was stopped just before the initial amount passed the filter, which allowed collecting the 2D-ink consisting of highly concentrated FL-BP/Sb material from the filter surface with a Teflon spatula. The 2D-inks were transferred to air and spread onto clean gold foils that were mounted on XPS sample holders. After exposure for several hours to air, the sample holders were introduced into the UHV system. Spectra were taken for the pristine samples and after heating in UHV in order to remove most of bmim-BF$_4$ by thermal evaporation (and partial decomposition as proven by XPS) to maximize P and Sb signal intensities. The heated samples were then exposed to air for about one day and measured again.

**General reaction procedure**. The corresponding nucleophile (0.1 mmol) and alkylating ester (0.15 mmol) were added under ambient conditions to a solution/dispersion of the corresponding catalyst (0.005 mmol for Brönsted acids and organic and inorganic bases, 0.02 for P, 0.01 mmol for Sb, and 1 mg for solids) in bmim-BF$_4$ (100 mg), placed in a 2 ml vial equipped with a magnetic stir bar. The vial was sealed and the resulting mixture was magnetically stirred at the required temperature for 4–20 h. Then, the reaction mixture was cooled and extracted with diethyl ether (1.5 ml). The extracts were analyzed by GC and GC–MS after adding dodecane (22.4 µl, 0.2 mmol) as an external standard, and the products were isolated by preparative thin-layer chromatography.

For kinetics, each point was taken from an individual reaction. For preparation purposes, scale is proportionally increased up to grams of starting material. For reuses, 2 mmol of starting materials are used and volatiles are removed from the extracted reaction mixture, under vacuum at room temperature for 15 min, prior to addition of fresh reactants. For longer catalyst lifetime, sonication after each reaction cycle of the IL mixture under inert atmosphere was carried out.

**Calorimetry**. 750 mg of IL-FL-BP (750 mg) was evacuated under vacuum for 30 min in a glass line, and a glass ampule was made in-situ, when still under vacuum. The ampule is submerged in a 1 M solution of benzyl alcohol in diethyl ether, inside the calorimetry apparatus, and then broken. Exchanged energy was measured at 30 °C during 2–4 h.

**Electrochemistry**. Voltammetric experiments were performed using a CH 660c equipment on FL-BP and FL-Sb diluted in the IL, after successive additions of BrOAc and $t$-BuOAc until 1 M concentration. Glassy carbon electrode (geometrical area 0.071 cm$^2$) was used as a working electrode, completing the three-electrode arrangement with a Pt mesh counter electrode and a Pt wire pseudo-reference electrode. A second series of experiments were performed in 0.10 M potassium phosphate buffer at pH 7.0 after forming a fine deposit of FL-BP and FL-Sb on glassy carbon electrode via transfer of 20 µL of each one of the above IL solutions, after 30 min of reaction, plus of 20 µL ethanol, evaporation at air and drying with a smooth paper tissue.

**Photocatalysis**. The photocatalytic water splitting is performed with a solar simulator light source (Newport®, Oriel Instruments, model 69921) equipped with a Xe lamp (1000 W) coupled with an AM1.5 filter that provides simulated concentrated sunlight in UV-visible range. 50 mg of FL-BP or FL-Sb were dispersed in 20 mL pure H$_2$O and then N$_2$ was purged into the reactor (quartz cell, 50 mL volume). Gas samples were periodically taken and analyzed in a micro-GC, using Ar as a standard.

## Data availability

The authors declare that all other data supporting the findings of this study are available within the paper and its Supplementary information files. The source data

underlying Figs. 1b, 2b–f, Figs. 1, 2h, I, and Supplementary Figs. 1–S9, Supplementary Figs. 11–S17, Supplementary Figs. 19, 20, Supplementary Figs. 22–24, Supplementary Figs. 28–30, Supplementary Figs. 32, Supplementary Figs. 34–40 are provided as a Source Data file. The Source Data file can be found in: Materials Cloud Archive [https://archive.materialscloud.org/2018.0021/v1]

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

## Acknowledgments

We thank the European Research Council (ERC Starting Grant 804110 to G.A., and ERC Advanced Grant 742145 B-PhosphoChem to A.H.) for financial support. The research leading to these results was partially funded by the European Union Seventh Framework Program under grant agreement No. 604391 Graphene Flagship. G.A. has received financial support through the Postdoctoral Junior Leader Fellowship Program from "la Caixa" Banking Foundation (LCF/BQ/PI18/11630018). G.A. thanks support by the Deutsche Forschungsgemeinschaft (DFG; FLAG-ERA AB694/2-1), the Generalitat Valenciana (SEJI/2018/034 grant) and the FAU (Emerging Talents Initiative grant #WS16-17_Nat_04). Financial support by MINECO through the Excellence Unit María de Maeztu (MDM-2015-0538), Severo Ochoa (SEV-2016-0683) and RETOS (CTQ2014-55178-R) program is acknowledged. M.A.R.-C. thanks MINECO for the concession of a FPU fellowship. We also thank the DFG (DFG-SFB 953 "Synthetic Carbon Allotropes", Project A1), the Interdisciplinary Center for Molecular Materials (ICMM), and the

Graduate School Molecular Science (GSMS) for financial support. Research at UCM sponsored by Spanish MINECO/FEDER grant MAT2015-066888-C3-3-R and ERC-PoC-2016 grant POLAR-EM. H.-P.S. thanks the European Research Council (ERC) under the European Union's Horizon 2020 research and innovation program for financial support, in the context of an Advanced Investigator Grant granted to him (Grant Agreement No. 693398-ILID). B.S.J.H. and S.S. acknowledge financial support by the DFG within the Cluster of Excellence "Engineering of Advanced Materials" (project EXC 315, Bridge Funding). F.M. acknowledges R. Ransom for very helpful discussions.

## Author contributions

G.A. and A.L.-P. conceived the research, designed the experiments, analyzed the data, supervised the project, and wrote the manuscript. V.L. and S.W. synthesized the samples. V.L., S.W., and G.A. performed AFM and Raman characterization; M.R.-C. and A.L.-P. performed the XRD experiments and the catalytic and kinetic studies; J.V.-M. contributed with the $^{31}P$ MAS-NMR measurements; A.D.-C. contributed with the electrochemical studies. M.V. performed STEM and EELS. F.H. and A.H. supervised the project. B.S.J.H, S.S., H.-P.S, and F.M. performed XPS characterization. All the authors discussed the results and contributed to writing the manuscript.

## Additional information

**Competing interests:** The authors declare no competing interests.

