## [Peer Review File · Nature Communications]

Reviewers' Comments:

Reviewer #1:

Remarks to the Author:

This manuscript describes the preparation of 2D nanosheets of BP and Sb exfoliated in the ionic liquid bmin-BF₄, which are active for the alkylation of soft nucleophiles with alkyl esters. These results are interesting. Therefore, I recommend to publish this work after revisions in the following:

1. The most important thing for the catalysis is the catalyst stability. Recycling for 3 times is not enough. In addition, authors should give the results for the solubility of the catalysts in the reactants and products, which is very important for the judgement of catalyst leaching.
2. The dispersion of catalyst active sites is very important. Therefore, authors should give average particle size distribution of the catalyst sizes.
3. Authors claimed the 2D nanosheets of the catalysts. Is it possible to obtain their XRD patterns?
4. After the deactivation, is it possible to regenerate the catalysts? If not, this work will lose its catalytic applications in the industrial processes.

Reviewer #2:

Remarks to the Author:

In my view, the revised manuscript addresses all prior concerns of the referees. The authors have suitably addressed the critiques on the catalytic aspects of their work. In particular, the authors have correctly noted the key scientific breakthrough of an organic reaction catalyzed by a 2D pnictogen. The particulars of the reaction are less important than the discovery of catalytic activity. Therefore, the manuscript is suitable for publication without further revision.

Reviewer #1 (Remarks to the Author):

1. The most important thing for the catalysis is the catalyst stability. Recycling for 3 times is not enough. In addition, authors should give the results for the solubility of the catalysts in the reactants and products, which is very important for the judgement of catalyst leaching.

We thank the Reviewer for the comments. Following his/her suggestion, and considering that flake agglomeration is the main cause of catalyst deactivation, we have repeated the reuses by carrying out the sonication of the ionic liquid mixture before each reaction cycle. The new results are included in Figure S24 of the Supporting Information and here, below, for the Reviewer's convenience. As it can be seen, the FL-BP catalyst prolongs its lifetime significantly, achieving a respectable 67% after 6 uses. In this way, the catalyst stability has doubled with a simple sonication procedure.

A new sentence has been included in the main text to highlight the Reviewer's concerns: "Following this rationale, and in order to increase the catalyst lifetime, sonication of the ionic liquid mixture was carried out before each reuse, and in this way, only a minor loss of final yield was observed after 6 reuses, thus doubling the catalytic performance without sonication." The new experimental procedure has also been included in the Methods section.

Regarding FL-BP solubility, we have washed the ionic liquid with the liquid reagents 1 and 2 and no signs of solubility for the black flakes can be observed. Please notice that, after ionic liquid removal, the BP severely agglomerates and losses most of its catalytic activity. These data reflects the good dispersion and stability of FL-BP imparted by the ionic liquid, essential to the catalysis.

2. The dispersion of catalyst active sites is very important. Therefore, authors should give average particle size distribution of the catalyst sizes.

These are elemental 2D materials exclusively composed of P or Sb atoms. The distribution of the catalyst sizes is already addressed in the main text (Pages 3 and 6, Figures 1 and 2 and supporting information) showing the average values of thickness and lateral dimensions of the catalysts. These data have been obtained by AFM measurements. Moreover, these samples have been characterized by SRM and XPS.

3. Authors claimed the 2D nanosheets of the catalysts. Is it possible to obtain their XRD patterns?

Following the Reviewer's concern, we have carried out now the XRD measurement of a 2D BP sample exfoliated in ionic liquid under nitrogen atmosphere, after ultracentrifugation, washing two times with THF, and then, after solvent evaporation (ionic liquids or any other solvent cannot be, in principle, present in our XRD instrumentation). The spectrum is shown in the new Figure S10 of the Supporting Information and below (left).

As it can be seen, the peaks and the d spacing values observed correspond to the typical 020, 040 and 060 planes of BP, without any sign of degradation nor oxidation (Nature Nanotechnology 2015, 10, 980; Scient. Rep. 2016, 6:34095). Indeed, when the sample is exposed to the ambient, the peak intensities rapidly decrease (see Figure in the right). These results, although do not show the 2D nanosheet spacing, confirm the BP nature of the FL-BP flakes and their rapid reactivity towards oxygen, in accordance with the high surface area.

The new experimental procedure for XRD measurements either under nitrogen or open air have been included in the manuscript, and a new sentence has been included in the main text: "X-ray diffraction (XRD) of a FL-BP sample, measured after washings with tetrahydrofurane (THF) under nitrogen atmosphere, ultracentrifugation and evaporation of the solvent, shows a spectrum consistent with BP, with the typical 020, 040 and 060 planes and without any sign of degradation nor oxidation, and when the sample was exposed to the ambient, the peak intensities rapidly decreased (Figure S10). These results infer the high surface area of the FL-BP synthesized here."

However, taken in account the Reviewer's concern about directly seeing the exfoliated BP nanosheets, we repeated the XRD without further eliminating the ionic liquid, in order to

preserve the exfoliation. As it can be seen in the Figure below, the reported, very weak band for exfoliated BP at aprox. 12° (Nature Nanotechnology 2015, 10, 980) can be inferred, despite the wide noise imparted by the ionic liquid.

Figure Ref.1. XRD spectrum of the BP exfoliated in ionic liquid under nitrogen atmosphere.

For further trying to measure the exfoliated BP by XRD, the exfoliated phosphorene solution in ionic liquid washed with THF was rapidly added on dehydrated high surface silica into the dry box. The impregnated solid was left to dry under a flow of nitrogen and then transferred to the XRD plate. The XRD spectrum shown below indicates that the BP maintains its nature when supported, although the amorphous silica band hides the minor bands associated to exfoliated BP.

Figure Ref.2. Top: XRD spectrum of the BP exfoliated in ionic liquid, washed with THF and impregnated onto high surface silica under nitrogen atmosphere. Bottom: silica blank.

Experimental procedure:

Samples were prepared through two different procedures A and B:

A) First, a suspension containing the BP prepared in ionic liquid, after ultracentrifugation and transferred to THF solution was deposited on a Si crystalline surface inside the dry box, the solvent was let to evaporate and then, the sample holder was covered with Kapton film to keep the inert atmosphere during the measurement. Another sample was prepared in a similar manner without Kapton film to let the sample exposed to the air.

B) The samples measured following the first procedure tend to aggregate, giving the signals of the bulk BP. In order to avoid the agglomeration of the sample, a methodology based on dispersion of the delaminated BP on a support was devised. SiO₂ powder (300 m²g⁻¹) was degassed and dried at 300 °C under vacuum during 3 hours, then put inside the dry box where 1 ml of the suspension of delaminated BP in THF was added and evaporated over night.

4. After the deactivation, is it possible to regenerate the catalysts? If not, this work will lose its catalytic applications in the industrial processes.

Please see answer to question 1. As it can be seen there, the deactivated catalyst is significantly regenerated after sonication, doubling its lifetime. The Reviewer's concern has been addressed with a new sentence in the main text.

Reviewer #2 (Remarks to the Author):

In my view, the revised manuscript addresses all prior concerns of the referees. The authors have suitably addressed the critiques on the catalytic aspects of their work. In particular, the authors have correctly noted the key scientific breakthrough of an organic reaction catalyzed by a 2D pnictogen. The particulars of the reaction are less important than the discovery of catalytic activity. Therefore, the manuscript is suitable for publication without further revision.

We thank the reviewer for his/her positive evaluation.

Reviewers' Comments:

Reviewer #1:

Remarks to the Author:

After the revisions, this work might be accepted for the publication.